# Cranial Base Synchondrosis Lacks PTHrP-Expressing Column-Forming Chondrocytes

**DOI:** 10.3390/ijms23147873

**Published:** 2022-07-17

**Authors:** Shawn A. Hallett, Annabelle Zhou, Curtis Herzog, Ariel Arbiv, Wanida Ono, Noriaki Ono

**Affiliations:** 1Department of Periodontics and Oral Medicine, University of Michigan School of Dentistry, Ann Arbor, MI 48109, USA; shallett@umich.edu (S.A.H.); abellez@umich.edu (A.Z.); arbiv@umich.edu (A.A.); 2Department of Oral and Maxillofacial Surgery, University of Michigan School of Dentistry, Ann Arbor, MI 48109, USA; crherzog@umich.edu; 3Department of Orthodontics, University of Texas Health Science Center at Houston School of Dentistry, Houston, TX 77054, USA; wanida.ono@uth.tmc.edu; 4Department of Diagnostic and Biomedical Sciences, University of Texas Health Science Center at Houston School of Dentistry, Houston, TX 77054, USA

**Keywords:** bone biology, cartilage, chondrocyte(s), craniofacial biology/genetics, developmental biology, growth/development, cranial base, PTHrP

## Abstract

The cranial base contains a special type of growth plate termed the synchondrosis, which functions as the growth center of the skull. The synchondrosis is composed of bidirectional opposite-facing layers of resting, proliferating, and hypertrophic chondrocytes, and lacks the secondary ossification center. In long bones, the resting zone of the epiphyseal growth plate houses a population of parathyroid hormone-related protein (PTHrP)-expressing chondrocytes that contribute to the formation of columnar chondrocytes. Whether PTHrP^+^ chondrocytes in the synchondrosis possess similar functions remains undefined. Using Pthrp-mCherry knock-in mice, we found that PTHrP^+^ chondrocytes predominantly occupied the lateral wedge-shaped area of the synchondrosis, unlike those in the femoral growth plate that reside in the resting zone within the epiphysis. In vivo cell-lineage analyses using a tamoxifen-inducible Pthrp-creER line revealed that PTHrP^+^ chondrocytes failed to establish columnar chondrocytes in the synchondrosis. Therefore, PTHrP^+^ chondrocytes in the synchondrosis do not possess column-forming capabilities, unlike those in the resting zone of the long bone growth plate. These findings support the importance of the secondary ossification center within the long bone epiphysis in establishing the stem cell niche for PTHrP^+^ chondrocytes, the absence of which may explain the lack of column-forming capabilities of PTHrP^+^ chondrocytes in the cranial base synchondrosis.

## 1. Introduction

The growth and development of the craniofacial skeleton is a complex process involving cells of multiple embryonic origins including the neural crest and the mesoderm [1]. The cranial base supports the central nervous system [2] and is composed of three cartilaginous synchondroses, including the spheno-occipital synchondrosis (SOS), intersphenoid synchondrosis (ISS), and spheno-ethmoidal synchondrosis (SES). The cranial base synchondrosis contains opposite-facing layers of resting, proliferating, and hypertrophic chondrocytes and lacks the secondary ossification center [3] (Figure 1). Malformations of the cranial base are associated with various types of genetic craniofacial disorders such as Crouzon, Pfeiffer, and Apert syndromes [4,5,6,7,8]. These conditions typically involve premature fusion of the synchondrosis and midfacial hypoplasia that leads to skeletal Class III malocclusion. Current treatment for these craniofacial skeletal disorders involves extensive craniofacial reconstructive surgery including maxillofacial advancement involving the Le Fort osteotomy followed by orthodontic treatment [9,10]. Given their genetic similarity to humans, we utilize several transgenic mouse models in this study to discern the regulation of the cranial base synchondroses.

The growth and development of the cranial base are incompletely understood [11,12,13,14], unlike those of the long bone that have been thoroughly studied [15,16]. Developmentally, the SOS is derived from both the mesoderm and the neural crest, whereas the ISS is solely derived from the neural crest [3]. It is generally assumed that the long bone growth plate and cranial base synchondrosis are formed and organized via similar mechanisms. However, this notion has been questioned in several instances [13]. In long bones, the secondary ossification center (SOC) separates the articular surface and growth plate, which functions to alleviate mechanical loads in the skeletons of amniotes [17]. The formation of the SOC within the epiphysis induces a subset of growth plate chondrocytes to acquire the capabilities for self-renewal and column-formation in long bones [18]. SOC formation coincides with the formation of skeletal stem cells expressing parathyroid hormone-related protein (PTHrP) within the resting zone [19]. Thus, the SOC may facilitate the formation of growth plate stem cells and their subsequent differentiation. However, the cranial base synchondrosis lacks the SOC. It is therefore unknown whether PTHrP^+^ chondrocytes similarly possess skeletal stem cell properties in the cranial base synchondrosis.

PTHrP is a constituent of the PTHrP—Indian hedgehog (Ihh) feedback loop that maintains chondrocyte proliferation and differentiation, functioning as an important regulator of endochondral bone growth [16]. PTHrP is expressed by round (fetal) or resting (postnatal) chondrocytes located at the top of the growth plate and delays the differentiation of proliferating chondrocytes. IHH is expressed by pre-hypertrophic chondrocytes and stimulates chondrocyte proliferation in the adjacent layer. IHH also enhances PTHrP expression in the round chondrocyte layer [20]. Mice lacking PTHrP display reduced proliferation of chondrocytes associated with premature hypertrophy and accelerated bone formation [21]. Similarly, in the cranial base, PTHrP-deficient mice display extensive premature hypertrophy of the synchondrosis leading to midfacial hypoplasia [22]. Thus, PTHrP is operative in both the long bone growth plate and cranial base synchondrosis.

In this study, we set out to investigate the contribution of PTHrP^+^ chondrocytes to the cranial base synchondrosis. We focused our analyses on the SOS, which is the last cartilaginous synchondrosis to ossify in humans [23,24]. Combined spatiotemporal characterization of *Pthrp* mRNA expression and Pthrp-mCherry reporter activities revealed that PTHrP-expressing cells are predominantly localized to the lateral wedge-shaped areas of the synchondrosis. Additionally, cell-lineage analyses using *Pthrp-creER* revealed that PTHrP^+^ cell-derived columnar chondrocytes are absent in the cranial base synchondrosis, in sharp contrast with the long bone growth plate wherein PTHrP^+^ chondrocytes in the resting zone contribute robustly to columnar chondrocytes. Therefore, PTHrP^+^ chondrocytes do not appear to function as skeletal stem cells in the cranial base synchondrosis, highlighting the divergent functions of PTHrP^+^ chondrocytes across different endochondral bones. These findings also suggest the importance of the SOC in establishing the resting zone stem cell niche within the growth plates, the absence of which may explain the lack of column-forming capabilities of PTHrP^+^ chondrocytes in the cranial base synchondrosis.

## 2. Results

### 2.1. PTHrP Expression Is Confined to the Lateral Wedge-Shaped Area of the Synchondrosis

First, we assessed the endogenous *Pthrp* mRNA expression patterns in the cranial base synchondrosis using RNAScope fluorescent in situ hybridization assays. At postnatal day 3 (P3), the spheno-occipital synchondrosis (SOS) was composed of three layers of round, flat, and hypertrophic chondrocytes (Figure 2A, top leftmost panel). The round chondrocytes in the central area were relatively homogenous, without distinct columns of chondrocytes being formed at this stage. This pattern continued at P6 (Figure 2A, top left center panel). However, at P9, columnar chondrocytes composed of elongated proliferating chondrocytes and hypertrophic chondrocytes appeared in the SOS (Figure 2A, bottom right and center panels, arrowheads). In the SOS, *Pthrp* mRNA was barely detectable at P3, P6, and P9 (Figure 2A). However, at P14, *Pthrp* became detectable in a wedge-shaped area on the lateral borders of the SOS adjacent to the surrounding peripheral connective tissue (Figure 2A, top right panel, dotted area).

We subsequently examined *Pthrp-mCherry* knock-in mice to define the PTHrP reporter activities in the cranial base synchondrosis. A PTHrP knock-in reporter transgene provides a more sensitive readout than in situ hybridization [25]. The contrast between the cranial base synchondrosis and the long bone growth plate is particularly instructive, as described below. In the femur, as we previously reported, Pthrp-mCherry^+^ cells were initially localized to the lateral portion of growth plates, adjacent to the groove of Ranvier at P3 and P6 [19] (Figure 2B). Following the formation of the SOC at P9, a new group of Pthrp-mCherry^+^ cells occupied the resting zone (Figure 2B). In contrast, in the cranial base synchondrosis, PTHrP-mCherry^+^ chondrocytes were primarily localized to the lateral portion of the synchondrosis, particularly within wedge-shaped areas adjacent to the resting zone (Figure 2C, yellow dotted lines). Although the SOS expanded superior-inferiorly at P14, PTHrP-mCherry^+^ cells were mostly confined to the wedge-shaped areas, whereas a small group of large round chondrocytes (reminiscent of hypertrophic chondrocytes) occupied the central portion of the resting zone of the SOS (Figure 2C, green dotted lines; enlarged in Figure 2E, arrowheads). Surrounding this central zone, a small number of Pthrp-mCherry^+^ flat chondrocytes were also formed at this stage (Figure 2C, arrows). Moreover, we observed inconsistent localization of Pthrp-mCherry^+^ cells in the surrounding marrow space; however, this could be attributed to autofluorescence in bone marrow stromal cells. Additionally, we injected a thymidine analogue, EdU, shortly before sacrifice to evaluate cell proliferation. Although incorporated randomly at P3, EdU was predominantly incorporated by proliferating chondrocytes at P6 and later time points. In contrast, Pthrp-mCherry^+^ cells in the wedge-shaped area were mostly devoid of EdU incorporation.

We further quantified the number of Pthrp-mCherry^+^ (PTHrP^+^), EdU^+^ (proliferating), and Pthrp-mCherry^+^EdU^+^ (PTHrP^+^ proliferating) cells (Figure 2D) in the cranial base synchondrosis and the central portion of the femur growth plate, which are similar in size. Pthrp-mCherry^+^ cells were found in similar abundance and increased progressively from P0 to P9, after which time they decreased in quantity (Figure 2D, upper). EdU^+^ cells increased steadily in the femoral growth plate from P0 to P14 (Figure 2D, middle, red line). Conversely in the SOS, EdU^+^ cells increased transiently at P6, then decreased thereafter. In the SOS, Pthrp-mCherry^+^EdU^+^ cells reached the highest number at P3, then decreased and reached a plateau thereafter (Figure 2D, lower, blue line). In contrast, in the femoral growth plate, the number of Pthrp-mCherry^+^EdU^+^ cells were relatively low until P6, and progressively increased thereafter. Thus, PTHrP^+^ chondrocytes are slow-cycling and relatively resistant to EdU incorporation in the femoral growth plate prior to SOC formation.

Together, these findings demonstrate that PTHrP^+^ chondrocytes in the cranial base synchondrosis are largely non-proliferative and predominantly confined to the lateral wedge-shaped area. Therefore, PTHrP^+^ chondrocytes in the cranial base synchondrosis may be different from those in the long bone growth plate.

### 2.2. PTHrP^+^ Chondrocytes Lack Column-Forming Capabilities in the Cranial Base Synchondrosis

We subsequently asked whether PTHrP^+^ chondrocytes in the cranial base synchondrosis include a population of column-forming cells. For this purpose, we utilized a tamoxifen-inducible Pthrp-creER line to trace the fate of PTHrP^+^ chondrocytes. First, to determine the location of the PTHrP^+^ cells that can be lineage-marked by this transgene, we pulsed Pthrp-creER; R26RtdTomato (PTHrP^CE^-tdTomato^+^) mice with a single dose of tamoxifen at sequential postnatal time points of P6, P9, P12, P15, and P25, followed by a subsequent chase for 72 h (Appendix A). This short-chase protocol is expected to mark PTHrP^+^ chondrocytes as tdTomato^+^ as a result of *cre-loxP* recombination. We observed similar numbers of PTHrP^CE^-tdTomato^+^ chondrocytes in the SOS when pulsed between P6 and P15, which declined substantially at P25 (Appendix A). We selected P6 to mark the PTHrP^+^ chondrocytes and analyzed their cell fates in our subsequent analyses.

For the following experiments, Pthrp-creER; R26RtdTomato mice were pulsed at P6 and analyzed up to 90 days of the chase, respectively, to assess the column-forming capability of PTHrP^+^ chondrocytes (Figure 3A, right). Only a minimal number of tdTomato^+^ cells were found across all layers of the SOS after the chase (Figure 3B). In contrast, consistent with our previous study, tdTomato^+^ cells were abundantly present in the resting zone of the femoral growth plate following the short chase at P9 and P12, which further differentiated into columnar chondrocytes (Figure 3C).

Quantification revealed that only a small number of tdTomato^+^ chondrocytes were present in the resting zone of the SOS at all time points, in contrast to abundant tdTomato^+^ cells in the resting zone of the femoral growth plate (Figure 3D). Moreover, essentially no tdTomato^+^ proliferating chondrocytes existed in the SOS at all time points, in contrast to abundant tdTomato^+^ cells in the proliferating zone of the femoral growth plate at P21, P36, and P96 (Figure 3E). Thus, a small number of PTHrP^+^ chondrocytes present in the resting zone of the cranial base synchondrosis do not differentiate into proliferating chondrocytes and form columns in the cranial base synchondrosis.

## 3. Discussion

In this study, we described the uniquely limited contribution of PTHrP^+^ chondrocytes to the cranial base synchondrosis. The two major findings of our study are (1) *PTHrP* expression (based on mRNAs and reporter activities) is spatiotemporally restricted to the lateral wedge-shaped areas of the cranial base synchondrosis, and (2) PTHrP^+^ chondrocytes do not function as skeletal stem cells without forming columnar chondrocytes in the synchondrosis (Figure 4). Our findings shed light on a functional difference of PTHrP^+^ chondrocytes between the cranial base synchondrosis and the long bone growth plate.

The difference between the cranial base synchondrosis and long bone growth plate has been previously reported, highlighting the well-known phenomenon that long bones possess a greater potential for longitudinal growth than that of the cranial base [26,27]. Additionally, although *PTHrP* expression domains appear to be different compared to the femur growth plate, the cell types within these structures possess similar hierarchies of resting, proliferating, and hypertrophic chondrocytes, further highlighting that PTHrP may have different functions in different endochondral bones. We previously reported that Pthrp-creER labels a population of resting chondrocytes with skeletal stem cell capabilities in the resting zone of the postnatal growth plate [19]. Interestingly, PTHrP^+^ resting chondrocytes acquire their stemness following the formation of the secondary ossification center (SOC) within the epiphysis. These PTHrP^+^ resting chondrocytes give rise to proliferating and hypertrophic chondrocytes and eventually to osteoblasts and bone marrow stromal cells in the metaphyseal marrow compartment. An intriguing possibility is that the cranial base lacks the capability to establish the stem cell niche within the resting zone. A unique feature of the long bone epiphysis is the presence of the SOC. Although detailed mechanisms underlying this event remain to be elucidated, our findings suggest that the formation of SOCs may be a key modulator facilitating sustained longitudinal growth of long bones.

One alternate hypothesis is that the decreased potential for growth during postnatal development in the synchondrosis could be the direct result of low PTHrP expression, thereby leading to decreased PTHrP—Ihh feedback and diminished chondrocyte proliferation. Furthermore, PTHrP’s role as a paracrine/autocrine regulator in skeletal development has long been established [28]. PTHrP promotes bone formation through local paracrine/autocrine-related mechanisms, and a PTHrP analog (abaloparatide) has been utilized as a therapeutic agent for osteoporosis treatment [29,30]. Yet, these observations have only been reported in long bones. Thus, it remains unknown whether PTHrP possesses similar paracrine functions in the cranial base. This is of critical importance, as growth deficiencies in the cranial base manifest as midfacial hypoplasia. Stimulation of cranial base growth using anabolic therapies may be a unique alternative to surgery for the repair of craniofacial skeletal malformations.

PTHrP^+^ chondrocytes in the wedge-shaped area appear to lack the capability to function as skeletal stem cells in the cranial base synchondrosis, emphasizing the differences in skeletal stem cell populations contributing to the growth of different endochondral bones. Although this study has not yet elucidated the presence of a skeletal stem cell population in the cranial base synchondrosis, future investigations using combinatorial lineage-tracing analyses and functional gene knockout approaches will highlight putative stem cell populations in diverse endochondral bones.

## 4. Materials and Methods

### 4.1. Mice

*Pthrp-creER, Pthrp-mCherry* [19], and *Rosa26-CAGG-lsl-tdTomato-WPRE (Ai14)* reporter [31] mice have been described previously. All animal experimental procedures were reviewed and approved by the Institutional Animal Care and Use Committees (IACUC) of the University of Michigan, protocol 9496, and the University of Texas Health Science Center at Houston, protocol AWC-21-0070. All experimental procedures followed the ARRIVE 2.0 guidelines for preclinical animal studies. Animals were not sorted for sex. Animal husbandry is provided by the staff of the Unit for Laboratory Animal Medicine (ULAM) under the guidance of supervisors who are certified as Animal Technologists by the American Association for Laboratory Animal Science (AALAS). Veterinary care is provided by ULAM faculty members and veterinary residents. The University of Michigan is fully accredited by the American Association for Accreditation of Laboratory Animal Care (AALAC) and the animal care and use program conforms to the standards in “the Guide for the Care and Use of Laboratory Animals,” DHEW Pub. No. (NIH)78-23, Revised 1978. This includes regular surveillance of animal facilities, a review of all funded projects for the humane use of animals, and the appropriate use of surgical anesthesia, analgesics, and tranquilizers. The University of Michigan has filed an assurance statement of these matters with the Office of Protection from Research Risk at the NIH.

### 4.2. Histology

Skulls and femurs were dissected under a stereomicroscope (Nikon SMZ-800, Tokyo, Japan) to remove soft tissues and fixed in 4% paraformaldehyde (PFA)/PBS for a proper period, ranging from 3 h to overnight at 4 °C, then decalcified in 15% EDTA for a proper period, typically ranging from 3 h to 14 days. Decalcified samples were cryoprotected in 30% sucrose/PBS solutions and then in 30% sucrose/PBS:OCT (1:1) solutions, each at least overnight at 4 °C. Samples were embedded in the sagittal plane in an OCT compound (Tissue-Tek, Sakura, Torrance, CA, USA) under a stereomicroscope and transferred on a sheet of dry ice to solidify the compound. Embedded samples were cryosectioned at 14–50 µm using a cryostat (Leica CM1850, Wetzlar, Germany) and adhered to positively charged glass slides (ColorFrost Plus, Fisher, Waltham, MA, USA). Cryosections were stored at −20 °C (quantification) or −80 °C (in situ hybridization and immunofluorescence) in freezers until use. Sections were postfixed in 4% PFA/PBS for 15 min at room temperature.

For reporter and lineage tracing assays, 20 µm serial sections were collected through the SOS and the femoral growth plate. For RNAScope in situ hybridization assays, microdissected cranial bases and long bone epiphyses were fixed for 24 h in 4% PFA/PBS and sectioned at 12 µm. Sections were further incubated with DAPI (4′,6-diamidino-2-phenylindole, 5 µg/mL, Invitrogen D1306, Waltham, MA, USA) to stain nuclei prior to imaging. Stained samples were mounted in TBS with No.1.5 coverslips (Fisher, Waltham, MA, USA).

### 4.3. RNAScope In Situ Hybridization

In situ hybridization was performed with RNAscope 2.5 Multiplex Fluorescent V2 Assay (Advanced Cell Diagnostics [Newark, CA, USA] 323100) using the following probes: Pthlh (456521) according to the manufacturer’s fixed frozen tissue protocol. Probes were diluted to 1:500 concentration using Opal 520 reagent (Akoya Biosciences [NC1601877], Marlborough, MA, USA).

### 4.4. Tamoxifen

Tamoxifen (Sigma T5648, St. Louis, MO, USA) was mixed with 100% ethanol until completely dissolved. Subsequently, a proper volume of sunflower seed oil (Sigma S5007) was added to the tamoxifen–ethanol mixture and rigorously mixed. The tamoxifen–ethanol–oil mixture was incubated at 60 °C in a chemical hood until the ethanol evaporated completely. The tamoxifen–oil mixture was stored at room temperature until use. Mice at 6 days of age received a single dose of 0.25 mg tamoxifen intraperitoneally for in vivo lineage-tracing experiments.

### 4.5. Edu Cell Proliferation and Label-Retaining Assay

To evaluate cell proliferation, 5-ethynyl-2′-deoxyuridine (EdU) (Invitrogen A10044) dissolved in PBS was administered to mice at indicated postnatal days. Click-iT Imaging Kit with Alexa Flour 488-azide (Invitrogen, C10337) was used to detect EdU in cryosections. EdU was pulsed once at all time points (50 μg) three hours before sacrifice.

### 4.6. Imaging and Cell Quantification

Images were captured by an automated inverted fluorescence microscope with a structured illumination system (Zeiss Axio Observer Z1 with ApoTome.2 system) and Zen 2 (blue edition) software. The filter settings used were FL Filter Set 34 (Ex. 390/22, Em. 460/50 nm), Set 38 HE (Ex. 470/40, Em. 525/50 nm), Set 43 HE (Ex. 550/25, Em. 605/70 nm), Set 50 (Ex. 640/30, Em. 690/50 nm), and Set 63 HE (Ex. 572/25, Em. 629/62 nm). The objectives used were Plan-Apochromat 10×/0.45, EC Plan-Neofluar 20×/0.50, EC Plan-Neofluar 40×/0.75, and Plan-Apochromat 63×/1.40. Images were typically tile-scanned with a motorized stage, Z-stacked and reconstructed by a maximum intensity projection (MIP) function. Differential interference contrast (DIC) was used for objectives higher than 10×. Regions of interest for quantification of mCherry^+^, tdTomato^+^, and EdU^+^ cells include all layers of the SOS and the central portion of the growth plate resting and proliferating zones. The number of mCherry^+^, tdTomato^+^, and EdU^+^ cells were counted by two individuals manually or using ImageJ image analysis software by single-blinded methods to ensure unbiased data interpretation.

### 4.7. Statistical Analysis

Results are presented as mean values ± s.d. Statistical evaluation was conducted based on one-way ANOVA followed by the Mann–Whitney U-test. A *p*-value <0.05 was considered significant.

## Figures and Tables

**Figure 1 ijms-23-07873-f001:**
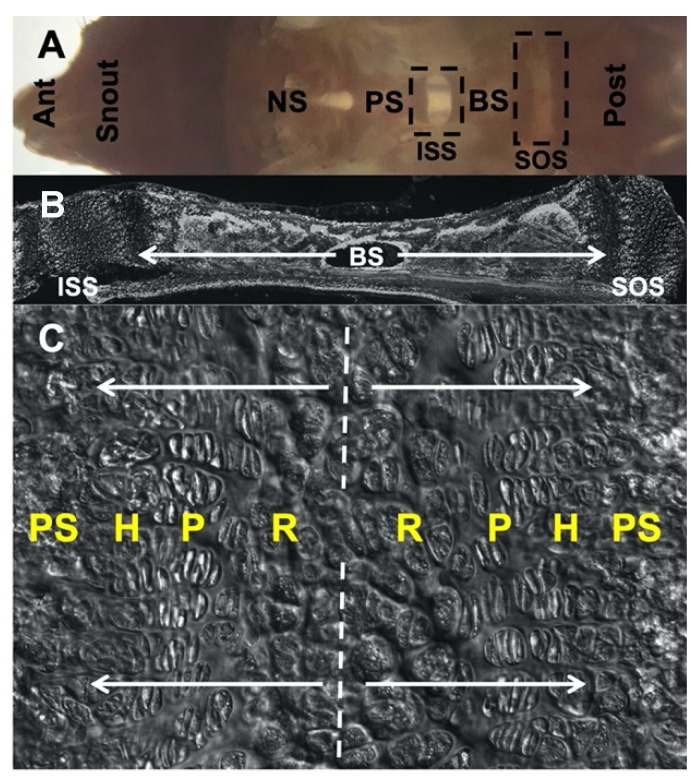
Morphology of cranial base and spheno-occipital synchondrosis (SOS), C57BL/6 mouse at postnatal day 28. (**A**) Gross morphology of dissected cranial base positioned in an anteroposterior manner and taken from the dorsal view following the removal of the brain and cranial vault. (**B**) Sagittal section of inter-sphenoid synchondrosis (ISS), basisphenoid bone and SOS. (**C**) Magnified image highlighting bidirectional arrangement of chondrocyte layers in SOS. Ant: anterior, NS: nasal septum, PS: pre-sphenoid, BS: basisphenoid, Post: posterior, ISS: inter-sphenoid synchondrosis, SOS: spheno-occipital synchondrosis, R: resting zone, P: proliferating zone, H: hypertrophic zone, PS: primary spongiosa.

**Figure 2 ijms-23-07873-f002:**
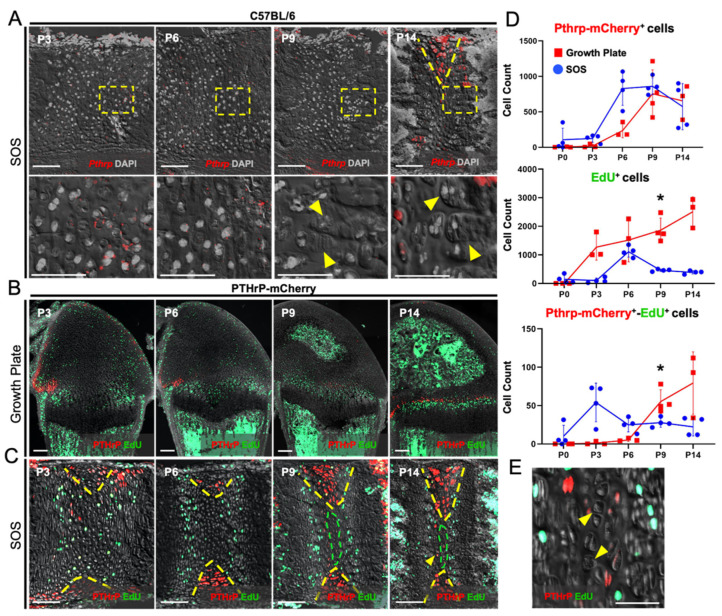
PTHrP expression is confined to the lateral wedge-shaped area of the synchondrosis. (**A**, Upper panels) *RNAScope* in situ hybridization assays for endogenous *Pthrp* mRNAs in early postnatal development. (**A**, Lower panels) 40× magnified images of transition zones between resting, proliferating, and hypertrophic chondrocytes. (**B**,**C**) Pthrp-mCherry knock-in reporter activities and EdU-labeling in femoral growth plate (**B**) and spheno-occipital synchondrosis (SOS) (**C**). Red: Pthrp-Opal 570 (**A**), Pthrp-mCherry (**B**,**C**), gray: DAPI (**A**) DIC (**B**,**C**). Scale bars: 100 μm. (**D**) Quantification of Pthrp-mCherry^+^ (Upper), EdU^+^ (Middle) and Pthrp-mCherry^+^-EdU^+^ (Bottom) chondrocytes. *n* = 3/4 of each group/timepoint. * *p* < 0.05, one-way ANOVA followed by the Mann–Whitney U test. Data are present as the mean ± SD. (**E**) 40× magnification of central hypertrophic zone in P14 SOS. (**A**,**C**) Yellow dashed wedges: PTHrP-mCherry expression domain restricted to the lateral wedge-shaped areas, (**A**, lower center panel) arrowheads: proliferating columns, (**A**, lower right panel) arrowheads: hypertrophic chondrocytes, (**C**) green dashed lines: presumptive central hypertrophic zone, (**E**) arrowheads: presumptive central hypertrophic chondrocytes.

**Figure 3 ijms-23-07873-f003:**
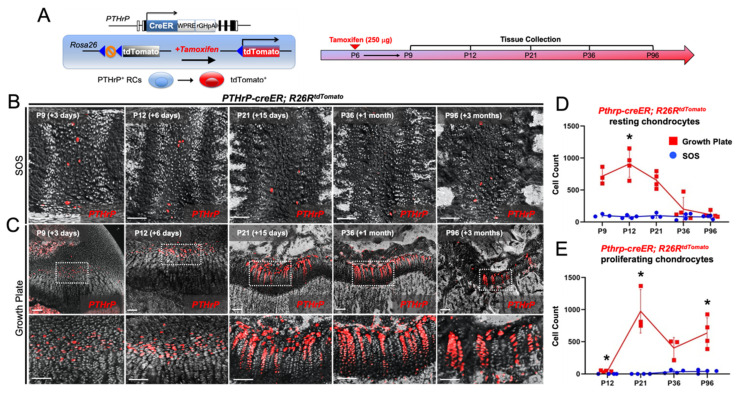
PTHrP^+^ chondrocytes lack column-forming capabilities in the cranial base synchondrosis. (**A**) Diagram of *Pthpr-creER; R26R-tdTomato* lineage-tracing model (Left) and tissue collection time points (Right). (**B**,**C**) Cell-lineage analysis of Pthrp-creER; R26RtdTomato^+^ (tamoxifen at P6) chondrocytes in the cranial base synchondrosis (**B**) and the femoral growth plate (**C**) following 3 (P9), 6 (P12), and 15 days (P21) and 1 month (P36) and 3 months (P96) of the chase. (**C**, Upper panels) Overview of femoral growth plate. (**C**, Lower panels) 40× magnified views of Pthrp-creER; R26R-tdTomato^+^ chondrocytes in the resting zone of the femur growth plate. Red: tdTomato, gray: DIC. Scale bars: 100 μm. (**D**) Quantification of Pthrp-creER; R26R-tdTomato^+^ resting chondrocytes. (**E**) Quantification of Pthrp-creER; R26R-tdTomato^+^ proliferating chondrocytes. *n* = 3/4 of each group/timepoint. * *p* < 0.05, one-way ANOVA followed by the Mann–Whitney U test. All data are present as the mean ± S.D.

**Figure 4 ijms-23-07873-f004:**
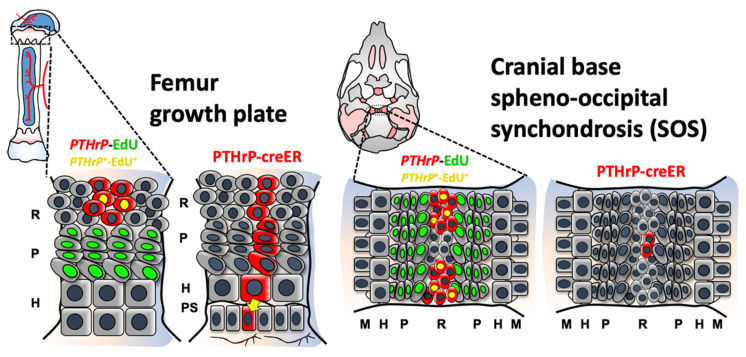
Cranial base synchondrosis lacks PTHrP^+^ column-forming chondrocytes in the resting zone. PTHrP^+^ chondrocytes are present both in the femoral growth plate and cranial base synchondrosis, which are generally characterized by their slow-cycling nature. Our cell-lineage analysis using *Pthrp-creER; R26R^tdTomato^* mice reveals that PTHrP^+^ chondrocytes in the synchondrosis fail to form columnar chondrocytes. Thus, PTHrP^+^ chondrocytes do not possess the characteristics of skeletal stem cells in the synchondrosis, highlighting functional differences between skeletal stem cells and their niches in two classes of endochondral bones—the cranial base and long bone. Red cells: mCherry^+^, tdTomato^+^, Green cells: EdU^+^, Yellow cells: mCherry^+^; EdU^+^. R: resting zone, P: proliferating zone, H: hypertrophic zone, PS: primary spongiosa, M: metaphyseal bone.

## Data Availability

Not applicable.

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
