# Peer review of "Cranial Base Synchondrosis Lacks PTHrP-Expressing Column-Forming Chondrocytes"

_ijms, 2022, doi:10.3390/ijms23147873_

Round 1
Reviewer 1 Report
The article results open the possibility that the mechanisms involved in the growth and development of the cranial base synchondrosis and the long bone growth plate are different. The current work shows a differential PTHrP expression pattern observed in the synchondrosis from the cranial base compared to that of the long bone growth plate. In addition, the authors point out that PTHrP+ chondrocytes across different endochondral bones have divergent functions. this conclusion is based on a analysis using a tamoxifen-inducible Pthrp-creER line to trace the fate of the PTHrP+ chondrocytes in a (PTHrP-tdTomato) mice. In contrast with the long bone, they do not observe a clear tdTomato+ signal in the chondrocytes of the cranial base synchondrosis.
Minor comments
1.-The description of the quantification of cell numbers is long and unnecessary in the result section.
2.-The discussion and conclusion of the paper are short and poor. The long bones possess a greater potential for longitudinal growth that the cranial base, which could explain the decrease in the PTHrp protein in the cranial base synchondrosis. Therefore, it could be interesting to include a discussion of the role of the hormonal mechanisms involved in the growth and development of the bone in the area.
3.-The authors do not clarify whether their work proposes that the chondrocytes that barely express PTHrP protein in the cranial base synchondrosis are different cell to the long bone chondrocytes.
Author Response
Reviewer 1
- The article results open the possibility that the mechanisms involved in the growth and development of the cranial base synchondrosis and the long bone growth plate are different. The current work shows a differential PTHrP expression pattern observed in the synchondrosis from the cranial base compared to that of the long bone growth plate. In addition, the authors point out that PTHrP+ chondrocytes across different endochondral bones have divergent functions. This conclusion is based on a analysis using a tamoxifen-inducible Pthrp-creER line to trace the fate of the PTHrP+ chondrocytes in a (PTHrP-tdTomato) mice. In contrast with the long bone, they do not observe a clear tdTomato+ signal in the chondrocytes of the cranial base synchondrosis.
Thank you very much for your overall assessment on our manuscript.
Minor comments
- The description of the quantification of cell numbers is long and unnecessary in the result section.
Thank you for this important suggestion. We have removed this extraneous information from the results section (Lines 153, 155, 159, 189, 200, 203).
2. The discussion and conclusion of the paper are short and poor. The long bones possess a greater potential for longitudinal growth that the cranial base, which could explain the decrease in the PTHrp protein in the cranial base synchondrosis. Therefore, it could be interesting to include a discussion of the role of the hormonal mechanisms involved in the growth and development of the bone in the area.
Thank you for this important comment. We have updated the discussion to reflect how PTHrP levels in the long bone growth plate and cranial base synchondrosis may explain the significant differences observed between the growth potentials of the two tissues (Lines 229-232). We have also updated the discussion to include a section on the role of PTHrP to regulate skeletal growth (Lines 244-255).
3. The authors do not clarify whether their work proposes that the chondrocytes that barely express PTHrP protein in the cranial base synchondrosis are different cell to the long bone chondrocytes.
Thank you for this important comment. We have clarified that although these chondrocytes express low levels of PTHrP in the cranial base synchondrosis and do not possess multi-lineage differentiation trajectories, they appear to be similar cells morphologically to those present in the femur growth plate (Lines 197-199).
Reviewer 2 Report
This is a very focused manuscript that describes the role of PTHrP as a growth factor and biomarker of the chondrocyte proliferation during development in two very different growing areas in the skeleton: the cranial base synchondrosis and the femoral growthplate. It illustrates the different behaviour of skeletogenesis depending on the developmental origin and anatomic characteristics of the bones. I do not see any major point to be addressed, but it would be interesting to address the suggestions below to increase the clarity of the manuscript.
Line 100 – Replace allele by transgene. Allele is not appropriate here.
I observed some double spaces and words separated (i.e. hyper trophic). Also some plural to singular inconsistencies (Pthrp-mCherry+ cells in the wedge-shaped area was mostly…). Please review orthography and grammar
Figure 1
Please use a anatomic cartoon similar to the one in figure 3 to anatomically define the areas shown in figure 1. This is not a skeletal-specific journal and specially the SOS is a site that most of the readers will not be familiarized to.
Describe in the legend the meaning of the arrowheads and lines in the figure
What is the yellow nuclei meaning in Figure 3? Please clarify
Why are the PTHrP-ER cartoons different (thinner) than the PTHrP-EdU cartoons? If there is not a specific reason to make them different, is it possible to use the same ones? That would ease the point authors are trying to make with this figure in my opinion.
En the ventral view of the cranial base the inter-sphenoid synchondrosis is also highlighted by an ellipse but not mentioned. Also, it says Cranial base synchondrosis (SOS) but SOS stands for spheno-occipital synchondrosis. Please correct these discrepancies. If the cartilage cartoon intends to illustrate both, it looks like it only illustrates the SOS.
Author Response
Reviewer 2
- This is a very focused manuscript that describes the role of PTHrP as a growth factor and biomarker of the chondrocyte proliferation during development in two very different growing areas in the skeleton: the cranial base synchondrosis and the femoral growth plate. It illustrates the different behaviour of skeletogenesis depending on the developmental origin and anatomic characteristics of the bones. I do not see any major point to be addressed, but it would be interesting to address the suggestions below to increase the clarity of the manuscript.
Thank you very much for your overall assessment on our manuscript.
Minor comments
- Line 100 – Replace allele by transgene. Allele is not appropriate here.
We apologize for this oversight in vocabulary. We have updated the text as requested (Line 128).
- I observed some double spaces and words separated (i.e. hyper trophic). Also some plural to singular inconsistencies (Pthrp-mCherry+ cells in the wedge-shaped area was mostly…). Please review orthography and grammar
We apologize for this oversight in orthography and grammar. We have updated the text as requested and ensured that all other sections are grammatically and orthographically accurate (Lines 139, 148).
- Figure 1. Please use a anatomic cartoon similar to the one in figure 3 to anatomically define the areas shown in figure 1. This is not a skeletal-specific journal and specially the SOS is a site that most of the readers will not be familiarized to. Describe in the legend the meaning of the arrowheads and lines in the figure
We thank you for this suggestion. We have generated a new summary Figure 1 to detail the anatomic location of the cranial base synchondroses and cellular composition. We have also updated the legend of Figure 1 (new Figure 2) to include descriptions for the arrowheads and lines (Lines 175-178).
- What is the yellow nuclei meaning in Figure 3? Please clarify
We apologize for this confusion. We have updated the figure and its legend (Line 271) to clarify that yellow nuclei indicate that these cells are positive for PTHrP (red) and EdU (green).
- Why are the PTHrP-ER cartoons different (thinner) than the PTHrP-EdU cartoons? If there is not a specific reason to make them different, is it possible to use the same ones? That would ease the point authors are trying to make with this figure in my opinion.
We apologize for the confusion. We have updated Figure 3 (new Figure 4), as requested, to include same-sized cartoons for both PTHrP-EdU and PTHrP-creER summaries.
- En the ventral view of the cranial base the inter-sphenoid synchondrosis is also highlighted by an ellipse but not mentioned. Also, it says Cranial base synchondrosis (SOS) but SOS stands for spheno-occipital synchondrosis. Please correct these discrepancies. If the cartilage cartoon intends to illustrate both, it looks like it only illustrates the SOS.
We apologize for the oversight. We have updated Figure 3 (new Figure 4), as requested.